# Identification and phylogenetic analysis of the genus *Syringa* based on chloroplast genomic DNA barcoding

**Ruihong Yao**[1☯]**, Runfang Guo**[2☯]**, Yuguang Liu**[1]**, Ziqian Kou**[1]**, Baosheng Shi**[1]*

**1** College of Landscape Architecture and Tourism, Hebei Agricultural University, Baoding, P. R. China,
**2** Department of Bioengineering, Hebei Agricultural University, Baoding, P. R. China

☯ These authors contributed equally to this work.
* baoshengshi@hebau.edu.cn

## Abstract

DNA barcoding is a supplementary tool in plant systematics that is extensively used to resolve species-level controversies. This study assesses the significance of using two DNA barcoding loci (e.g., *psbA-trnH* and *trnC-petN*) in distinguishing 33 plant samples of the genus *Syringa*. Results showed that the average genetic distance K2P of *psbA-trnH* DNA marker was 0.0521, which is much higher than that of *trnC-petN*, which is 0.0171. A neighbor-joining phylogenetic tree based on *psbA-trnH* and *trnC-petN* indicated that the identification rate of *psbA-trnH* and *trnC-petN* alone were 75% and 62.5%, respectively. The barcode combination of *psbA-trnH*+*trnC-petN* could identify 33 samples of the genus *Syringa* accurately and effectively with an identification rate of 87.5%. The 33 *Syringa* samples were divided into four groups: Group I is series *Syringa* represented by *Syringa oblata*; Group II is series *Villosae* represented by *Syringa villosa*; Group III is series *Pubescentes* represented by *Syringa meyeri*; and Group IV is section *Ligustrina* represented by *Syringa reticulata* subsp. *pekinensis*. These research results provided strong evidence that the combinatorial barcode of *psbA-trnH*+*trnC-petN* had high-efficiency identification ability and application prospects in species of the genus *Syringa*.

**Data Availability Statement:** All relevant data are within the paper and its Supporting Information files.

**Funding:** This work was supported by the Key R & D projects of Hebei Province, China

## Introduction

DNA barcodes enable the rapid and accurate identification of species using short, standardized DNA regions as species tags [1]. In addition to assigning specimens to known species, DNA barcoding will accelerate the pace of species discovery by allowing taxonomists to sort specimens rapidly and by highlighting divergent taxa that may represent new species [2]. DNA barcoding had been widely used in various biological fields because of its advantages of high sensitivity, accuracy, and objectivity [3–6]. One of the major challenges faced by barcoding is the ability to resolve sister species within a large geographical range. Consortium for the Barcode of Life (CBOL) recommended the use of two plastid loci (e.g., *matK* and *rbcL*) as the standard plant DNA barcode loci [7]. A large number of experiments had been conducted using these two markers in different taxa and species. However, the identification results were

(No.19226367D). The funders had no role in study design, data collection and analysis, decision to publish, or preparation of the manuscript.

**Competing interests:** The authors have declared that no competing interests exist.

unsatisfactory. Chase emphasized that the universality and identification effect of *matK* primers were not ideal [8]. Sass found that the *rbcL* often used for phylogenetic analysis across large groups of plants did not usually contain enough variability to identify individual species [9]. Increasing number of studies had shown that a system based on any one or small number of chloroplast genes will fail to differentiate taxonomic groups with extremely low amount of plastid variations while it will be effective in other groups [10,11]. Therefore, some scholars suggested that the screening of plant barcodes should not only focus on a single fragment but must be supplemented with additional fragment as required, and a combination of multiple fragment markers should be used [12]. Kress and Erickson combined the non-coding *trnH-psbA* spacer region, and the use of a portion of the coding *rbcL* gene as a two-locus global land plant barcode that provides the necessary universality and species discrimination is recommended [13]. Lahaye reported that the combination of *matK* to *trnH-psbA* and *psbK-psbI* could slightly increase its performance in identifying species [14]. Ho Viet identified 21 jewel orchids by *rbcL+matK* [15]. Meanwhile, Bhagya Chandrasekara found that *rbcL+matK+trnH-psbA* could still not completely solve the phylogenetic problem of *Cinnamomum* [16]. Therefore, for the species identification of different taxa, effective barcoding and their combination schemes, which can be used as supplementary markers for DNA barcoding, must be developed.

*psbA-trnH* and *trnC-petN* are chloroplast intergenic spacer sequences that are neither restricted by function nor affected by selection. Moreover, these two loci for the species level exhibited considerable genetic variability and divergence, ease of amplification, short sequence length, conserved flanking sites for developing universal primers, and ease of alignment and species relationship analysis [17]. Literature reported that *psbA-trnH* had successfully identified aquatic freshwater plants and the authenticity of herbal medicines accurately and effectively [18,19]. Niu sequenced *psbA-trnH* and 8 other chloroplast loci of 16 individuals of *Triplostegia* that represented the entire distribution range of both species recognized [20]. Similarly, *trnC-petN* showed high identification potential in *Triticum* plants [21], and Liu revealed the phylogenetic relationships and biogeographic diversification history of *Cissus*, which used *trnH-psbA* and *trnC-petN* markers [22].

The genus *Syringa* (family Oleaceae) are mainly distributed in southeast Europe, Japan, China, Afghanistan and North Korea. Approximately 27 wild species of the genus *Syringa* have been described, and most of which are native to China [23]. However, disputes about the infrageneric classification and relationships of the genus *Syringa* exist, and a comprehensive taxonomic system has not yet been established. The classification standard proposed by Zhang and Qiu that divided the genus *Syringa* into 2 sections and 4 series, including section *Syringa* and section *Ligustrina* is generally accepted. The section *Syringa* can be divided into series *Syringa*, series *Pinnatifoliae*, series *Pubescentes*, and series *Villosae* [24]. At present, some new varieties of the genus *Syringa* are constantly appearing in the market, but the classification standards are different, and the genetic relationship is uncertain. For example, on the question of species or subspecies of *S. wolfii*, it was classified as species in Flora Reipublicae Popularis Sinicae [24], but Chen pointed out that *S. wolfii* should be a subspecies of *S. villosa* [25]. In addition, no reports on the genetic relationship of *S.* 'Si Ji Lan', *S.* 'Zhan Mu Shi', and *S.* 'Xiang Ya Duan' are presented. Thus, solving these problems through morphological classification is challenging. Therefore, the main objective of this paper is to select gene fragments with multiple mutation loci according to the chloroplast genome sequence of the genus *Syringa*, identify various species of the genus *Syringa* by using sequence-specific markers, and develop DNA barcodes.

## Materials and methods

### Sample collection and DNA extraction

A total of 33 samples of the genus *Syringa* and 2 outgroup genera (Table 1) were collected from the garden nursery of Hebei Agricultural University and Beijing Botanical Garden in April–May 2021. The fresh leaves of the plant were placed in −80˚C fridge. The genomic DNA was extracted from leaves by using PlantGen DNA Kit (CWBIO). The quantification and purity of the extracted DNA were measured using NanoDrop 2000 (Thermo Scientific) and 1.2% agarose gel electrophoresis.

### PCR amplification

Based on the complete chloroplast genomes of five species of the genus *Syringa* in NCBI, intergenic spacer or intron regions with high variation were selected, and all the primers were designed by Primer primer 5.0 (Table 2). High-quality template DNA was used for PCR amplification (T100™ Thermal Cycler, BioRad). PCR reaction for *psbA-trnH* and *trnC-petN* was carried out in a total volume of 50 μL that contains 2 μL genomic DNA template, 3 μL of each primer, 25 μL 2 × Taq PCR MasterMix, 17 μL double distilled deionized water. The reaction conditions were initial denaturation at 94˚C for 2 min, subsequently 32 cycles starting with 94˚C denaturation for 30 s, annealing for 30 s, followed with a final extension at 72˚C for 45 s, followed by 72˚C for 8 min. The PCR products were detected by 1.2% agarose gel electrophoresis, and the bidirectional sequencing was completed by BGI (Beijing BGI Company).

### Data analysis

The sequencing results were aligned and spliced by using SeqMan software (DNAStar). The sequence data were further utilized to analyze the AT and GC contents and SNPs for each species using BioEdit. The software MEGA X was used to compare the obtained sequences and

**Table 1. Information of the plant materials.**

| No. | Taxonomic group | Species | Source | No. | Taxonomic group | Species | Source |
|---|---|---|---|---|---|---|---|
| 1 | Series *Syringa* | *S. vulgaris* 'Macroflora' | BeiJing | 18 | Series *Villosae* | *S. villosa* | BaoDing |
| 2 | | *S. vulgaris* 'Alba Plena' | BeiJing | 19 | | *S. josikaea* | BeiJing |
| 3 | | *S.* × *chinensis* | BeiJing | 20 | | *S. emodi* | BeiJing |
| 4 | | *S.* × *chinensis* 'Saugeana' | BaoDing | 21 | | *S.* 'Zhan Mu Shi' | BaoDing |
| 5 | | *S. oblata* var. *affinis* | BaoDing | 22 | Series *Pubescentes* | *S. pubescens* subsp. *patula* | BeiJing |
| 6 | | *S. oblata* | BaoDing | 23 | | *S. pubescens* subsp. *microphylla* | BaoDing |
| 7 | | *S. oblata* 'Ziyun' | BaoDing | 24 | | *S. pubescens* subsp. *microphylla* 'Superba' | BeiJing |
| 8 | | *S. oblata* subsp. *dilatata* | BaoDing | 25 | | *S. microphylla* 'Superba' | BaoDing |
| 9 | | *S.* × *hyacinthiflora* 'Luo Lan Zi' | BaoDing | 26 | | *S. meyeri* | BeiJing |
| 10 | | *S.* × *hyacinthiflora* 'Asessippi' | BaoDing | 27 | | *S. meyeri* 'Palibin' | BaoDing |
| 11 | | *S.* × *hyacinthiflora* 'Blanche Sweet' | BaoDing | 28 | | *S.* 'Si Ji Lan' | BaoDing |
| 12 | | *S.* × *hyacinthiflora* 'Mount Baker' | BaoDing | 29 | Section *Ligustrina* | *S. reticulata* subsp. *pekinensis* | BeiJing |
| 13 | Series *Villosae* | *S.* × *prestoniae* 'James Macfarlane' | BaoDing | 30 | | *S.* 'Jinyuan' | BeiJing |
| 14 | | *S.* × *prestoniae* 'Minuet' | BaoDing | 31 | | *S. reticulata* subsp. *amurensis* | BaoDing |
| 15 | | *S. tomentella* | BeiJing | 32 | | *S. reticulata* | BeiJing |
| 16 | | *S. wolfii* | BeiJing | 33 | | *S.* 'Xiang Ya Duan' | BaoDing |
| 17 | | *S. sweginzowii* | BeiJing | 34 | Outgroup | *Forsythia suspensa* | BaoDing |
| | | | | 35 | | *Ligustrum lucidum* | BaoDing |

**Table 2. PCR primer information.**

| Gen Bank | Accession no. | Region | Primer sequence (5'→3') | Tm/°C | PCR products size/bp |
|---|---|---|---|---|---|
| *Syringa wolfii* | NC049090 | *psbA-trnH* | F: GTTATGCATGAACGTAATGCTC | 56 | 336–518 |
| *Syringa vulgaris* voucher G. Besnard | NC036987 | | R: CGCGCATGGTGGATTCACAATCC | | |
| *Syringa yunnanensis* voucher | NC042468 | *trnC-petN* | F: TTTTTCCCCAGTTCAAATCCG | 54 | 778–804 |
| *Syringa pinnatifolia* | MG917095 | | R: GACTACCATTAAAGCAGCCCA | | |
| *Syringa persica* cv. *Laciniata* | NC042280 | | | | |

analyze the loci of variation. Average interspecific and intraspecific distances were calculated by using a Kimura 2-parameter (K2P) distance model. A neighbor-joining (NJ) phylogenetic tree on the sequences was performed using the software MEGA X with 1000 bootstrap replications to check the support rate of each fulcrum. In addition, data were also used to develop DNA barcodes for each species by using online DNA Barcode Generator (QR barcode) software (http://biorad-ads.com/DNABarcodeWeb/), and the *psbA-trnH* and *trnC-petN* sequences of the genus *Syringa* were transformed into two-dimensional images using the QR barcode approach (https://www.the-qrcode-generator.com).

## Results

### Sequence characteristics

The specific DNA fragments of all tested species of the genus *Syringa* were successfully amplified by using *psbA-trnH* and *trnC-petN* primers, the lengths of the amplified products were in the ranges of 336–518 bp and 778–804 bp, and the average lengths were 465 bp and 785 bp (Fig 1). Similarly, DNA sequencing also suggested that *psbA-trnH* and *trnC-petN* generated high-quality amplicons. *psbA-trnH* and *trnC-petN* achieved amplifying and sequencing efficiencies of 100%. In the amplicons of *psbA-trnH*, the average nucleotide composition of AT and GC for species of the genus *Syringa* was 71.28% and 28.72%, respectively. In *trnC-petN*, the average AT and GC were 61.76% and 38.16%, respectively. Moreover, all the sequences of amplicons were aligned with those sequences published on NCBI. The consistency of *psbA-trnH* was 81.45%–84.81%, and that of *trnC-petN* was 96.67%– 97.23%.

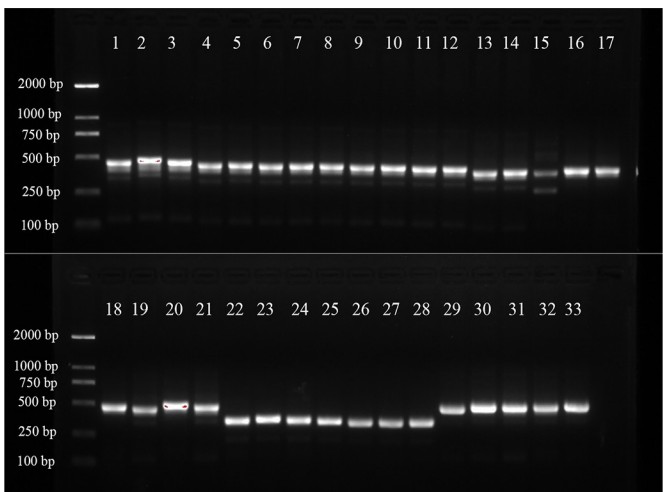
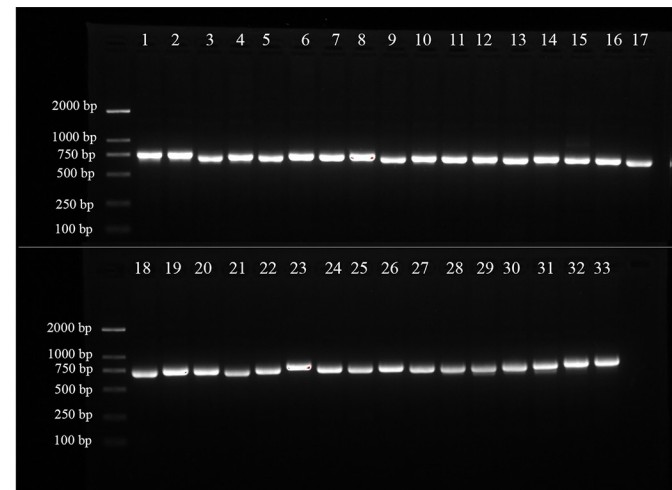

**Fig 1. PCR amplified products using *psbA-trnH* (left) and *trnC-petN* (right) primers.**

## Genetic distance between interspecific and intraspecific

The interspecific and intraspecific genetic distances of all samples were calculated by MEGA X software, and the results were shown in Fig 2. For *psbA-trnH*, the maximum and mean of K2P genetic distance in tested species of the genus *Syringa* were calculated as 0.1359 and 0.0521 ±0.0013, respectively. Similarly, for *trnC-petN*, the maximum and means of K2P genetic distance in the tested species were 0.0438 and 0.0171±0.0005, respectively. The distance of the *psbA-trnH* marker has been increased because it is considered a high potential barcoding region for the systematic study in plant evolution.

## Analysis of variant sites and barcodes

The results showed 91 variable sites (V), 73 parsimony-informative sites (Pi), 18 singleton sites (S) in *psbA-trnH*, and 45 variable sites (V), 41 parsimony-informative sites (Pi), and 4 singleton sites (S) in *trnC-petN* (S1 Table).

Unique barcodes with *psbA-trnH* 508 bp and *trnC-petN* 182 bp and 330 bp were highly conserved in series *Syringa*. Other species in the series *Syringa* had their own unique barcodes, except for sharing one barcode of *S. × hyacinthiflora* 'Asessippi', *S. × hyacinthiflora* 'Blanche Sweet', and *S. × hyacinthiflora* 'Mount Baker'.

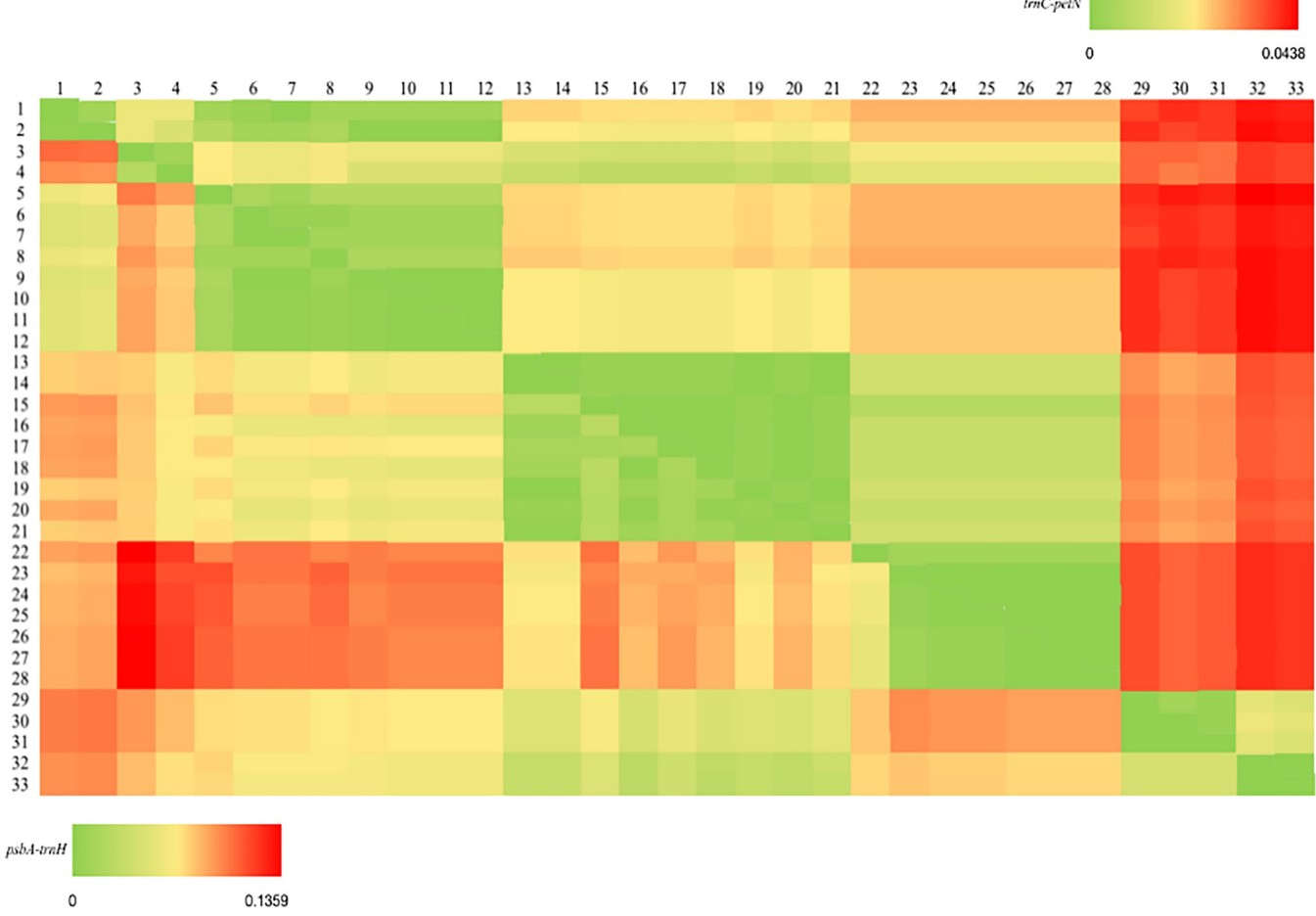

**Fig 2. Analysis of interspecific and intraspecific distance of the genus *Syringa* based on *psbA-trnH* (lower left) and *trnC-petN* (upper right).**

Series *Villosa* generated a highly conserved unique barcode in *psbA-trnH* 424 bp. *S. josikaea*, *S.* × *prestoniae* 'James Macfarlane' and *S.* × *prestoniae* 'Minuet' shared a single set of barcodes, whereas all others had shown characteristic barcode.

Series *Pubescentes* was identified by the absence of the barcode at *psbA-trnH* 162, 165, 166, 170–172, 176, 178, 182, 189, 231, 236, 260, 263, 272, 277, 286, 304, and 319 bp. Meanwhile, the presence of the highly conserved barcode in *trnC-petN* 381, 403, 431, 570 bp. *S. meyeri* shared barcodes with *S. meyeri* 'Palibin'. The others had unique DNA barcodes.

The highly conserved barcodes that identify section *Ligustrina* were *psbA-trnH* 172 bp and *trnC-petN* 401, 495, 497–499, 503, 505, 507, 511–513, 514, 535, and 802 bp. Each species of section *Ligustrina* had unique SNPs.

## Phylogenetic analysis for the *psbA-trnH* and *trnC-petN*

Phylogenetic tree based on *psbA-trnH* showed that *S.* × *chinensis* and *S.* × *chinensis* 'Saugeana', which belongs to series *Syringa*, was clustered inside the series *Villosae*, and the success rate of identification was 75% (Fig 3). However, in *trnC-petN*, a crossover between series *Villosae (S. tomentella)* and series *Pubescentes* was observed, and the identification success rate was 62.5% (Fig 4). The results showed that the two markers used alone could not distinguish all samples of the genus *Syringa*. Therefore, the phylogenetic tree was established by the two marker combinations. Meanwhile, series *Syringa*, series *Villosa*, series *Pubescentes*, and section *Ligustrina* formed independent branches, and the success rate of identification was 87.5% (Fig 5). Phylogenetic tree based on *psbA-trnH and trnC-petN* indicated that the 33 samples of the genus *Syringa* were divided into four groups: Group I is series *Syringa* represented by *S. oblata*; Group II is series *Villosae* represented by *S. villosa*; Group III is series *Pubescentes* represented by *S. meyeri*; and Group IV is section *Ligustrina* represented by *S. reticulata* subsp. *pekinensis*. The DNA barcodes of species of the genus *Syringa* were established using *psbA-trnH+trnC-petN* variable sites. The combination of two barcodes can distinguish the genus *Syringa*, and species of the genus *Syringa* information was captured by scanning the QR code image using a mobile terminal. Fig 6 only showed the QR code information of four representative the genus *Syringa* group based on *psbA-trnH+trnC-petN* sequence. The results showed that the combination of the barcodes *psbA-trnH* and *trnC-petN* were sufficient for classifying *Syringa* species.

## Discussion

Traditional morphological markers are greatly influenced by environmental factors, as well as the developmental stages of the plant. These markers failed to effectively distinguish some morphologically consistent species, which consist of *S. reticulata* subsp. *pekinensis* and *S. reticulata*. However, molecular markers had been extensively employed in species classification and identification because of their abundance and high polymorphism. Specifically, AFLP, SSR, ISSR, and other polymorphisms are identified by complying with changes in DNA length [26–28]. As indicated from the previous study of the authors, ISSR molecular markers were adopted to identify plants of the genus *Syringa* [29]. According to the results, gene exchange was reported between series *Pubescentes* and series *Villosae*, which was consistent with the results in Gao who used the germplasm characterization of different plants of the genus *Syringa* by applying AFLP markers [30]. Thus, neither of the two markers could accurately distinguish the two groups. As the sequencing technology had been leaping forward, the method of exploiting DNA sequence was recognized to be reliable and accurate in identifying species. DNA barcoding can accurately identify species by marking the sequence variation site. In this study, the built chloroplast DNA barcodes could identify 33 samples of the genus *Syringa* accurately.

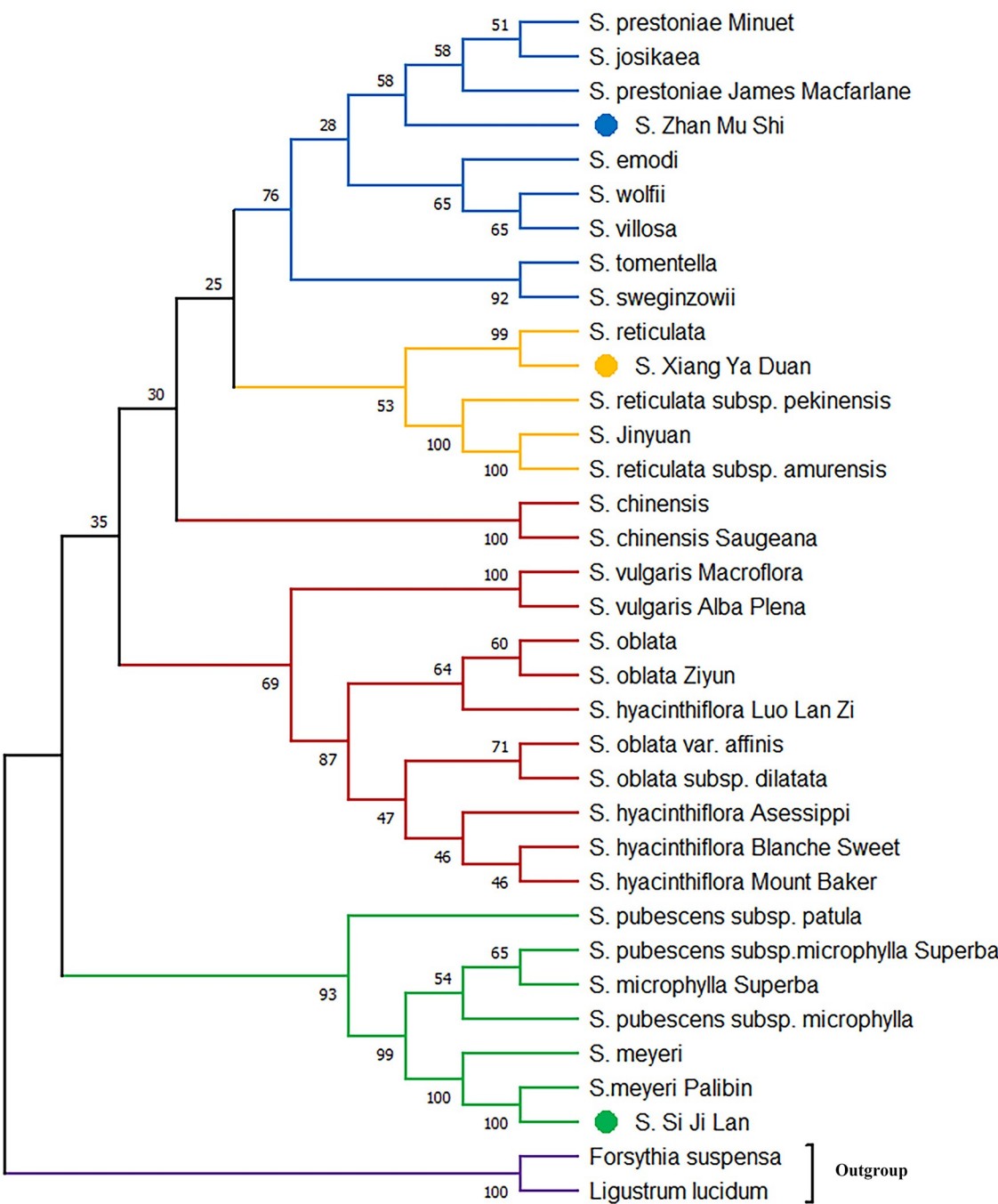

**Fig 3. Phylogenetic relationship among different species of the genus *Syringa* differentiated on the basis of *psbA-trnH* intergenic spacers.**

To establish the DNA barcode suitable for the identification of the plants of the genus *Syringa*, the complete chloroplast genomes of five species of the genus *Syringa* were first found on the NCBI. Then, sequence alignment was performed to screen the DNA fragments suitable for the barcode. Eventually, eight fragments with larger variations were determined as DNA barcode candidates. From the experience of other scholars, this study was carried out sequentially from the fragments with large variability [3,12], and *psbA-trnH* and *trnC-petN* had high

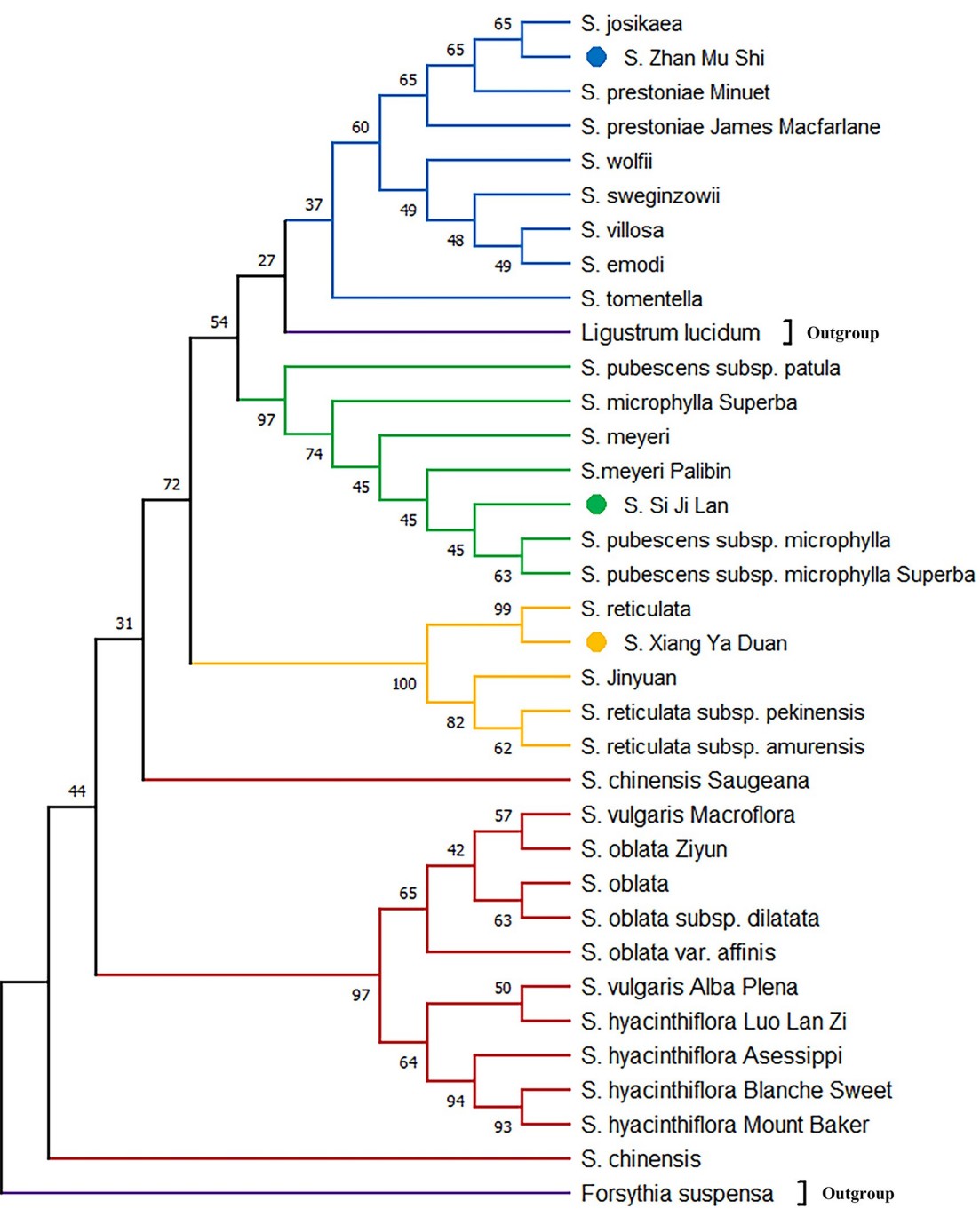

**Fig 4. Phylogenetic relationship among different species of the genus *Syringa* differentiated on the basis of *trnC-petN* intergenic spacers.**

and reliable identification abilities for the genus *Syringa*. The success rate of amplification and sequencing of *psbA-trnH* and *trnC-petN* fragment was 100%, and the identification rate of two marker combinations was 87.5%. The PCR amplification and sequencing success rates for *psbA-trnH* in 122 plant samples of Apocynaceae were 100% and 61%, and the identification efficiency at the species level is 82% [31]. A study used *trnC-petN* and other markers to

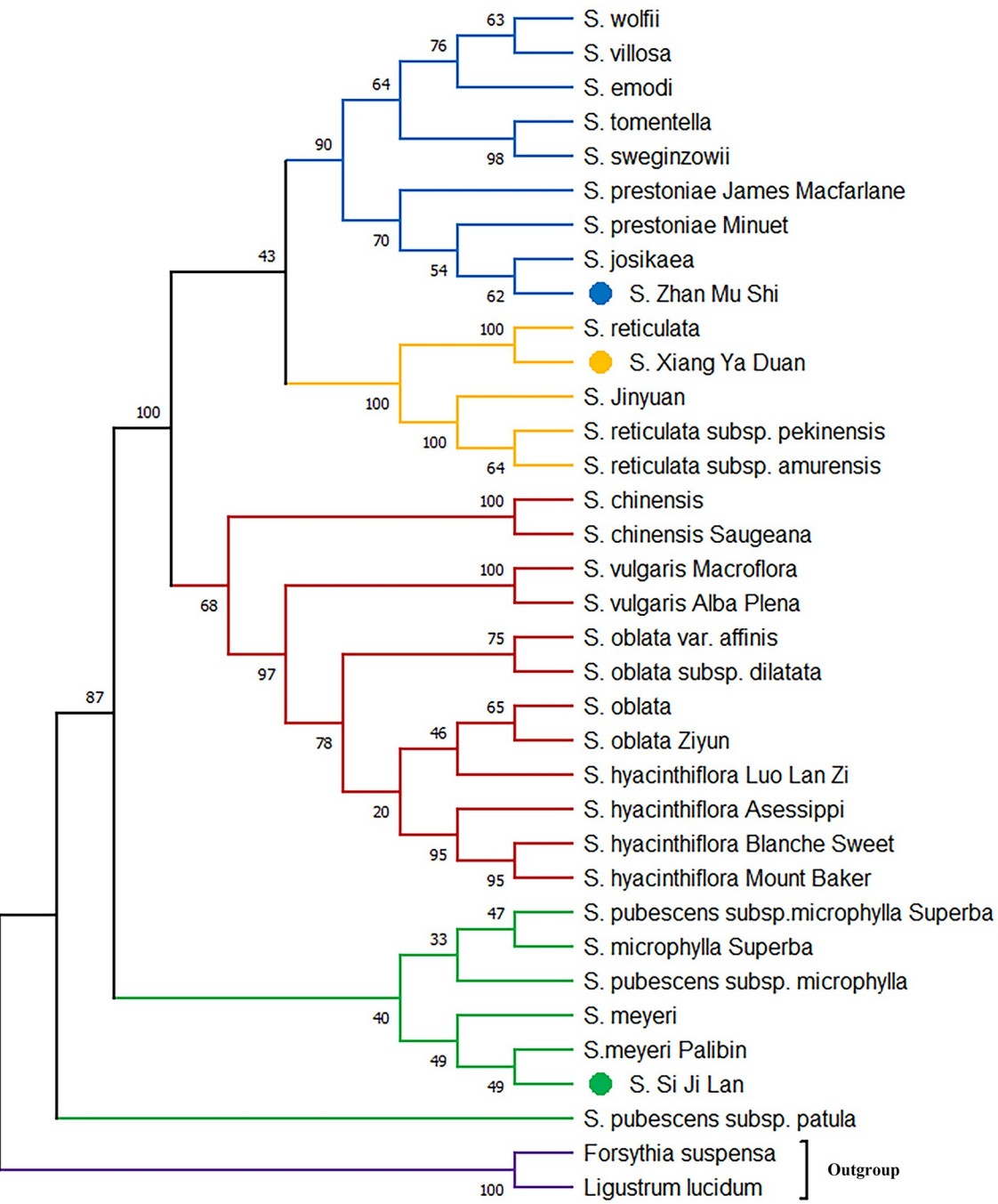

**Fig 5. Phylogenetic relationship among different species of the genus *Syringa* differentiated on the basis of *psbA-trnH* and *trnC-petN* intergenic spacers.**

construct the relationships and biogeographic diversification history of *Cissus* [22]. *psbA-trnH* and *trnC-petN* fragments can be used as DNA barcode options.

PCR amplification and sequencing results showed that *psbA-trnH* spans a large gene length (336–518 bp) because of the role of insertions/deletions in the evolution of the intergenic region in *psbA-trnH*, even among sister species [32]. As a result, the fragment length varied greatly among different plants. In this study, the *psbA-trnH* DNA length of series *Pubescentes*

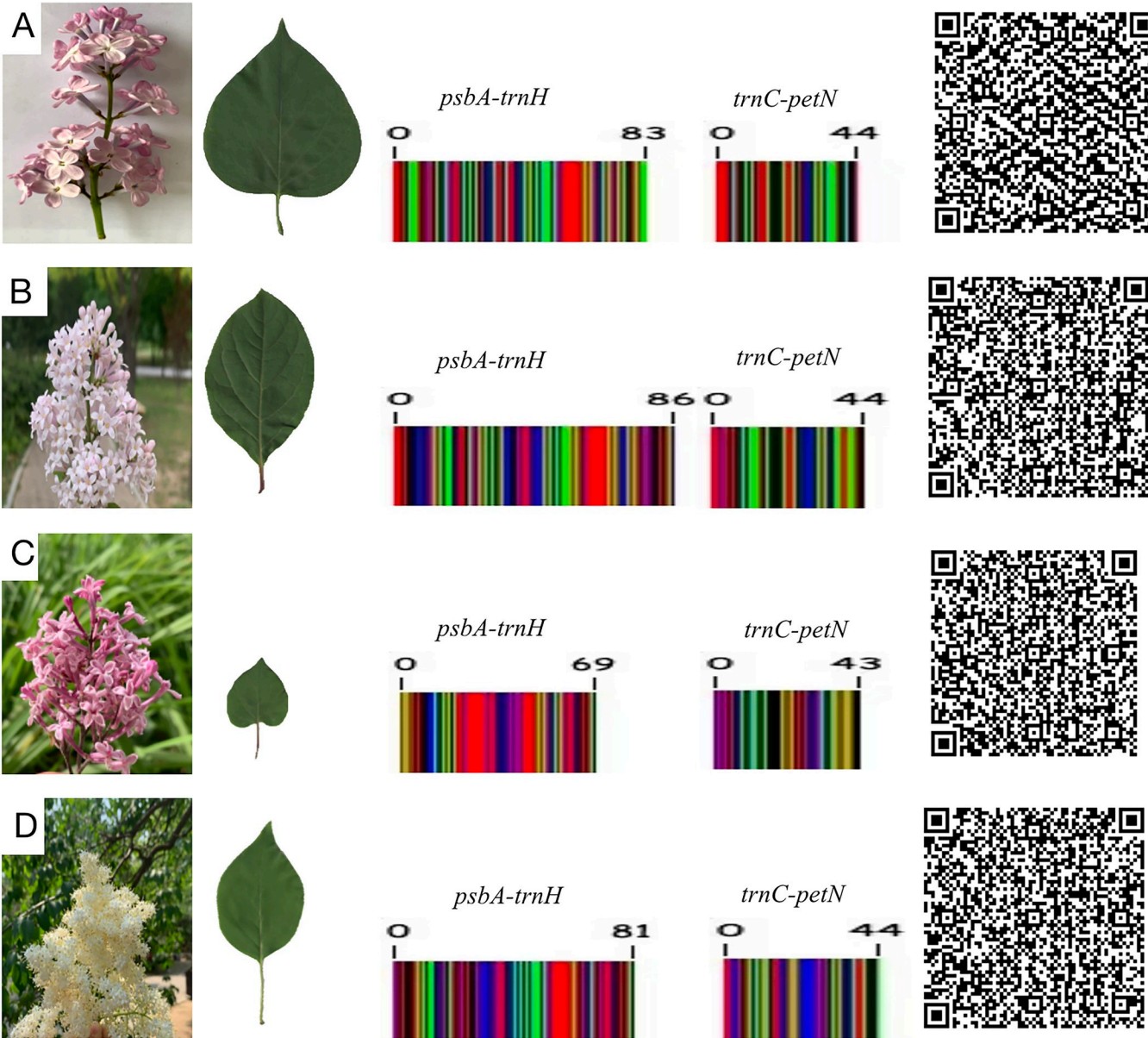

**Fig 6. Four species of the genus _Syringa_ morphology, DNA barcoding, and two-dimensional DNA barcoding image of _psbA-trnH_ and _trnC-petN_ sequences.** (A) _S. oblata_; (B) _S. villosa_; (C) _S. meyeri_; (D) _S. reticulata_ subsp. _pekinensis_. In the center-colored DNA image, the different colors represent various nucleotides (A T C G) and the numbers represent the lengths of the sequences that can be used in obtaining clear sequence information.

was significantly lower than those of other species. The length of _trnC-petN_ ranged from 778 bp to 804 bp, and the number of _S. reticulata_ and _S._ 'Xiang Ya Duan' was 803 bp, which was significantly higher than the 778 bp of _S. reticulata_ subsp. _pekinensis_ and _S. reticulata_ subsp. _amurensis_. The results also indicated that the degree of base variation was positively correlated with the distance of genetic relationship between species.

In this study, 33 samples were divided into four groups, namely, series _Syringa_, series _Villosae_, series _Pubescentes_, and section _Ligustrina_. In traditional morphological markers, section was divided according to the length of the corolla tube. Generally, the genus _Syringa_ could be divided into two types: section _Syringa_ and section _Ligustrina_ [24]. However, differences were

not observed in section at the chloroplast genome level. This finding was consistent with our results using ISSR molecular markers to examine the relationships of the genus *Syringa* [29]. This finding may be due to the weak linkage among these sequences or the molecular markers and corolla tube length traits used in the experiment. Yang conducted a correlation analysis between SSR markers and corolla traits and discovered that SO649 markers were linked to the length of the corolla tube. The transcriptome sequence of the SO649 marker was annotated as E3 ubiquitin-protein ligase, which was a B3 domain-containing protein. The B3 domain-containing protein is essential for stress responses and plant growth and development [23]. Therefore, the corolla tube length related genes were assumed to be located in the nuclear genome rather than in the chloroplast intergenic spacers. The anthers of *S. emodii* are longer than the corolla tube, which is consistent with the morphological classification of the section *Ligustrina* [33]. The results of Ki-Joong and Robert's cpDNA tree analysis revealed that *S. emodii* clustered in the series *Villosae* but not in the section *Ligustrina* [33], indicating a weak association between the corolla tube length and the chloroplast genes. In addition, the IPlant (http://www.IPlant.cn) and Chen proposed that *S. wolfii* was a subspecies of *S. villosa* [25]. In this study, *S. wolfii* was closely related to *S. villosa*, forming sister relationship. *S. wolfii* was identified as an independent species in the Flora Reipublicae Popularis Sinicae [24], and our study supported their view. Furthermore, the three unknown genetic relationship species were successfully identified by using *psbA-trnH* and *trnC-petN* fragments. The *S.* 'Si Ji Lan' was closely related to *S. meyeri*, the *S.* 'Zhan Mu Shi' was closely associated with *S. josikaea*, and the *S.* 'Xiang Ya Duan' was near *S. reticulata*. These two chloroplast genomic primers may provide sufficient molecular data for identifying closely related *Syringa* species.

The current study tested the effectiveness of these two fragments and their combination markers using a large number of experimental samples, and the identification efficiency of the combination markers below the species level was 85%. The result had shown that the chloroplast fragments *psbA-trnH* and *trnC-petN* could be used as identification barcodes of *Syringa* plants. Moreover, we developed QR codes based on the DNA sequence and established characteristic barcodes for each species.

## Supporting information

**S1 Table. Barcode of selected species of the genus *Syringa* based on variable regions of *psbA-trnH* and *trnC-petN* markers.**
(PDF)

**S1 Raw images.**
(PDF)

## Acknowledgments

The authors would like to thank Dr. Mengxin, Beijing Botanical Garden for her guidance throughout *Syringa* sample collection.

## Author Contributions

**Conceptualization:** Ruihong Yao, Runfang Guo, Baosheng Shi.

**Formal analysis:** Ruihong Yao, Runfang Guo.

**Funding acquisition:** Baosheng Shi.

**Investigation:** Ruihong Yao, Runfang Guo, Yuguang Liu, Ziqian Kou.

**Methodology:** Ruihong Yao, Runfang Guo.

**Resources:** Yuguang Liu.

**Supervision:** Ruihong Yao, Runfang Guo, Baosheng Shi.

**Writing – original draft:** Ruihong Yao.

**Writing – review & editing:** Runfang Guo.

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
