## [Decision Letter · Decision Letter 0]

19 May 2022

PONE-D-22-01886Identification and phylogenetic analysis of Syringa based on chloroplast genomic DNA barcodingPLOS ONE

Dear Dr. Shi,

Thank you for submitting your manuscript to PLOS ONE. After careful consideration, we feel that it has merit but does not fully meet PLOS ONE’s publication criteria as it currently stands. Therefore, we invite you to submit a revised version of the manuscript that addresses the points raised during the review process.

The manuscript has several major issues that prevent to be accepted for publication in the present form. Please see below my specific comments, under "Additional Editor Comments".

We look forward to receiving your revised manuscript.

Kind regards,

Branislav T. Šiler, Ph.D.

Academic Editor

PLOS ONE

Journal Requirements:

This work was supported by the Key R & D projects of Hebei Province, China (No.19226367D). 

This work was supported by the Key R & D projects of Hebei Province, China (No.19226367D). The authors would like to thank Dr. Mengxin for her assistance.  

This work was supported by the Key R & D projects of Hebei Province, China (No.19226367D). 

Additional Editor Comments:

The first impression is that it was written in a rather casual manner, i.e., it is abundant in typos and wrong punctuation, line and page numbers are lacking, vernacular phrases and vague sentences are met (e.g., second paragraph of Introduction: ", Niu et al. (2018)", "which used", the sentence that begins with "We obtained..." and the next sentence - are these the results of this study? If yes, they cannot stand in Introduction; if previously published, then citation should be provided); third paragraph: "The *Syringa*...plants" - vernacular expression; "...and other places" - scientifically unacceptable - please specify them; "Approximately 27 wild species of *Syringa*" - are there any non-wild species? Moreover, "of *Syringa*" is a colloquial expression - please write "species of the genus *Syringa*". What are "Sect and "*Ser*"? Are they "section" and "series"? Why they stand capitalized? Why is one italicized and other not? Please see e.g., https://www.newworldencyclopedia.org/entry/Taxon for proper writing. What is "Sort" in Table 1? What is the meaning of "inter- and intra- distance of *Syringa*" in the Fig.2 caption? Please also note that markers stated in the captions for Fig.3, Fig.4, and Fig.5 are not genetic markers but intergenic spacers. A meticulous editing by a native English speaker or a professional editing agency must be performed.

Regarding the results of the study and discussion upon them, the term "phylogenetic" should be avoided in this study since no dendrogram has been rooted. Please rectify the Discussion section in this sense.

Table 3 cannot stand in the main text, since being intermittent and too large. Please submit it as a supplementary material and rectify its referencing in the text.

Figures must be submitted as separate files in proper extensions and resolution. You may find more at https://journals.plos.org/plosone/s/figures. Please consult the PACE tool for checking those parameters.

Reviewers' comments:

Reviewer's Responses to Questions

**Comments to the Author**

1. Is the manuscript technically sound, and do the data support the conclusions?

Reviewer #1: Partly

2. Has the statistical analysis been performed appropriately and rigorously? 

Reviewer #1: N/A

3. Have the authors made all data underlying the findings in their manuscript fully available?

Reviewer #1: Yes

4. Is the manuscript presented in an intelligible fashion and written in standard English?

Reviewer #1: No

5. Review Comments to the Author

Reviewer #1: The paper evaluates the effectiveness of two well-known DNA sequence fragments at identifying plant specimens of the genus Syringa, and thus at their potential broader use as DNA barcodes within this taxon. The study is straightforward, although I am somewhat bothered by the fact that most taxa used therein are hybrids and cultivars.

While I am not familiar with the development of DNA barcodes, I found that part of the paper to be sound. However, I find it unsatisfactory to include "phylogenetic analysis/relationship" without any outgroup, especially given the fact that Syringa is paraphyletic. It is different if the goal is to molecularly characterize groups previously recognized with morphological characters (e.g., the sections), which was met with the barcode development.

The discussion about weak linkage or molecular markers and corolla tube should be further clarified and references should be added to that sentence.

With regards to format, the paper needs to be checked thoroughly, including tables and figures, for inconsistencies. For instance, subspecies should not be capitalized; names and words should be written correctly (e.g., "oblata" vs. "oblate"; "Syringa" vs. "Syring"; "section" vs. "Sect"); spaces should be added between words and after punctuations.

The paper can also benefit from further editing to improve the English language and to remove some awkwardness throughout the text. For instance, in the Introduction, "disputes about the under the genus classification and interspecific relationships …", why not just say "the infrageneric classification and relationships".

I deplore the lack of page and line numbers, which makes it more difficult to point to deficiencies accurately.

6. PLOS authors have the option to publish the peer review history of their article (what does this mean?). If published, this will include your full peer review and any attached files.

Reviewer #1: No

---

## [Author Response · Author response to Decision Letter 0]

22 Jun 2022

Dear Dr. Branislav T. Šiler:

Thank you very much for giving us an opportunity to revise our manuscript. We appreciate the editor and reviewers very much for their constructive comments and suggestions on our manuscript entitled “Identification and phylogenetic analysis of Syringa based on chloroplast genomic DNA barcoding” (ID: PONE-D-22-01886).

We have studied reviewers’ comments carefully. According to the reviewers’ detailed suggestions, we have made a careful revision on the original manuscript. All revised portions are marked in red in the “Revised Manuscript with Track Changes” which we would like to submit for your kind consideration.

Kind regards.

Ruihong Yao

E-mail: 337514209@qq.com

Corresponding author : Baosheng Shi

E-mail address: baoshengshi@hebau.edu.cn

Dear Dr. Branislav T. Šiler and reviewers:

Thank you for your letter and the reviewers’ comments on our manuscript entitled “Identification and phylogenetic analysis of Syringa based on chloroplast genomic DNA barcoding” (ID: PONE-D-22-01886). Those comments are very helpful for revising and improving our paper, as well as the important guiding significance to other research. We have studied the comments carefully and made corrections which we hope meet with approval. The main corrections are in the manuscript and the responds to the reviewers’ comments are as follows (the replies are highlighted in blue ).

Replies to the Editor Comments:

1.The first impression is that it was written in a rather casual manner, i.e., it is abundant in typos and wrong punctuation, line and page numbers are lacking, vernacular phrases and vague sentences are met (e.g., second paragraph of Introduction: “, Niu et al. (2018)”, “which used”, the sentence that begins with “We obtained...” and the next sentence - are these the results of this study? If yes, they cannot stand in Introduction; if previously published, then citation should be provided).

Response: We’d like to extend our apology for the trouble brought to you. We have checked the problem of format throughout the manuscript carefully, and made correction one by one. As for the format of citation in the reference, We will cite it again and add references to the citations. For instance, behind the “Niu et al. (2018)” in the Lines 80-82, I cited the 20th reference; behind “which used” in the Lines 84-85, I cited the 22th reference; “We obtained...” in the original paper is the content about the experiment in this chapter, and I have deleted it from the citation. 

 third paragraph: “The Syringa...plants” - vernacular expression; “...and other places” - scientifically unacceptable - please specify them; “Approximately 27 wild species of Syringa” - are there any non-wild species? Moreover, “of Syringa” is a colloquial expression - please write “species of the genus Syringa”.

Response: The words “The Syringa... plants” in the Line 86 of the third paragraph have been changed to “The genus Syringa...”. “... and other places” in Line 87 have been changed to “Afghanistan and North Korea”; “... wild species” in Line 87 referred to the words falling within the scope of description in the 23th reference, in which non-wild species refer to hybrids and cultivars. Every “of Syringa” in the paper has been changed to “species of the genus Syringa”.

 What are “Sect” and “Ser”? Are they “section” and “series”? Why they stand capitalized? Why is one italicized and other not? Please see e.g., https://www.newworldencyclopedia.org/entry/Taxon for proper writing. What is “Sort” in Table 1? 

Response: We would like to extend my gratitude to you for the website https://www.newworldencyclopedia.org/entry/Taxon offered. “Sect.” and “Ser.” refer to the abbreviation of “section” and “series” in this paper, and I have changed all the “Sect.” and “Ser.” to “section” and “series” in this paper. With regard to the problems about capital and small letters of the initial character and italics, I referred to A reference. The ways of writing are “section Syringa”, “section Ligustrina”, “series Pinnatifoliae”, and “series Pubescentes”. The word “Sort” in Table 1 under the Line 115 refers to the classification of “section” and “series”. It’s the problem related to our statement, and we have changed “Sort” to “Taxonomic group”.

(A). Li J, Zhang AD. Paraphyletic Syringa (Oleaceae): Evidence from Sequences of Nuclear Ribosomal DNA ITS and ETS Regions. Systematic Botany. 2002; 27(3): 592-597.

What is the meaning of “inter- and intra- distance of Syringa” in the Fig.2 caption? Please also note that markers stated in the captions for Fig.3, Fig.4, and Fig.5 are not genetic markers but intergenic spacers. A meticulous editing by a native English speaker or a professional editing agency must be performed.

Response: The phrase “Inter- and intra-distance of Syringa” in Line 172 in Fig.2 represents “interspecific and intraspecific distance of Syringa”, and the phrase has been changed to the complete spelling. The phrase “genetic markers” in Lines 219-226 in the Fig.3, Fig.4, and Fig.5 have been changed to “intergenic spacers”. We have polished the paper, and the proof is in the figure as follows. 

2. Regarding the results of the study and discussion upon them, the term “phylogenetic” should be avoided in this study since no dendrogram has been rooted. Please rectify the Discussion section in this sense.

Response: I’d like to extend my deep gratitude to you for pointing out the problems, because the problems pointed out by you indeed help perfect this paper. In the discussion, I have modified the use of the word “phylogenetic”. We have changed “phylogenetic relationships” in Line 266 in the second paragraph to “relationships”; We have changed the sentence “These two chloroplast genomic primers can be used to identify and clarify the phylogenetic relationships between the species and varieties of Syringa” in Lines 306-307 of the fourth paragraph to “These two chloroplast genomic primers may provide sufficient molecular data for identifying closely related Syringa species”.

3. Table 3 cannot stand in the main text, since being intermittent and too large. Please submit it as a supplementary material and rectify its referencing in the text.

Response: We have submitted Table 3 again as S1 Table, and have corrected its citation in the main body (Line 178).

4. Figures must be submitted as separate files in proper extensions and resolution. You may find more at https://journals.plos.org/plosone/s/figures. Please consult the PACE tool for checking those parameters.

Response: I’d like to extend my deep gratitude to you for the tools offered by you, and we have checked the figure again using the software PACE. We have submitted again and please check it.

Replies to the Reviewers’ Comments:

Reviewer #1:

1. The paper evaluates the effectiveness of two well-known DNA sequence fragments at identifying plant specimens of the genus Syringa, and thus at their potential broader use as DNA barcodes within this taxon. The study is straightforward, although I am somewhat bothered by the fact that most taxa used therein are hybrids and cultivars. 

Response: We’d like to convey our apology to you for the confusion brought to you. The establishment of phylogenetic relationship based on the informative sites of chloroplast genome sequence has been widely applied, which deals with the relationships among different orders, families, genera and even subspecies of angiosperm. For instance, the scholar established the phylogenetic tree by selecting 22 grape varieties in B reference.

But the varieties of cultivars, and its hybrids are high in number, with complicated genetic relationship. If they can be identified using DNA barcodes, the value of DNA barcodes can be better interpreted. For instance, the author studied the phylogenetic relationships and genomic compatibility were compared for 60 accessions of Syringa using chloroplast DNA (cpDNA) and nuclear ribosomal DNA (rDNA) markers in C reference. The plant material for this article includes Syringa cultivars and hybrids. The author studied the phylogenetic relationship between quinoa of different varieties in D reference. The author studied the phylogenetic relationship of peony using DNA barcodes in E reference. The research materials included 40 species, subspecies taxa or varieties of peony.

(B). Yang YM. A Phylogenetic Study of Vitis Based on Chloroplast Genomes. M.Ag. Thesis, Chinese Academy of Agricultural Sciences. 2019.

(C). Ki JK, Jansen RK. A chloroplast DNA phylogeny of lilacs (Syringa, Oleaceae): plastome groups show a strong correlation with crossing groups. American Journal of Botany. 1998; 85.

(D). Gao ZM. Complete chloroplast genomes of Chenopodium quinoa strains and phylogenetic relationship. M.Ag. Thesis, Shanxi University. 2021.

(E). Zhang JM, Wang JX, Xia T, et al. Application of DNA barcoding based on phylogenetic analysis in clarifying species problems of Paeonia. Scientia Sinica(Vitae). 2008; 38(12): 11.

2. While I am not familiar with the development of DNA barcodes, I found that part of the paper to be sound. However, I find it unsatisfactory to include “phylogenetic analysis/relationship” without any outgroup, especially given the fact that Syringa is paraphyletic. It is different if the goal is to molecularly characterize groups previously recognized with morphological characters (e.g., the sections), which was met with the barcode development. 

Response: I’d like to extend my deep gratitude to you for your suggestions, which allows this paper to be more perfect. Concerning the problem of outgroup, we have selected Forsythia suspensa and Ligustrum lucidum as outgroups, and conducted cluster analysis again, as shown in the figure as follows.

Fig 3. Phylogenetic relationship among different species of the genus Syringa differentiated on the basis of psbA-trnH intergenic spacers.

Fig 4. Phylogenetic relationship among different species of the genus Syringa differentiated on the basis of trnC-petN intergenic spacers.

Fig 5. Phylogenetic relationship among different species of the genus Syringa differentiated on the basis of psbA-trnH and trnC-petN intergenic spacers.

DNA barcoding has obvious advantage in the phylogenetic relationships of different varieties, as well as the determination of the relationship between different species and the homology of molecular characters. At the same time, it is applied in the species identification, molecular geography and research on the origin of species. Later, many scholars developed DNA barcoding for the identification of genetic relationships. For instance, nine endangered endemic plant species in SKP were selected to test the variable abilities of three different DNA barcodes by the authors, and it was found that the barcode sequences were efficient in finding the genetic relationships between the nine species in F reference.

(F). Amh A, Aa B, Fma C, et al. Phylogenetic Relationships and DNA Barcoding of Nine Endangered Medicinal Plant Species Endemic to Saint Katherine Protectorate. Saudi Journal of Biological Sciences. 2021; 28: 1919-1930.

3. The discussion about weak linkage or molecular markers and corolla tube should be further clarified and references should be added to that sentence. 

Response: (Lines 285-299) Thanks for your advice very much. The length of the corolla tube was used to split sections in traditional morphological markers. In general, the genus Syringa can be separated into two sections: Syringa and Ligustrina [24]. At the chloroplast genome level, however, no changes were found in sections. This outcome was in line with our findings from a study of Syringa genus connections using ISSR genetic markers [29]. Yang conducted a correlation analysis between SSR markers and corolla traits and discovered that SO649 markers were linked to the length of the corolla tube. The transcriptome sequence of the SO649 marker was annotated as E3 ubiquitin-protein ligase, which was a B3 domain-containing protein. The B3 domain-containing protein is essential for stress responses and plant growth and development [23]. Therefore, the corolla tube length related genes were assumed to be located in the nuclear genome rather than in the chloroplast intergenic spacers. The anthers of S. emodii are longer than the corolla tube, which is consistent with the morphological classification of the section Ligustrina [33]. The results of Ki-Joong and Robert's cpDNA tree analysis revealed that S. emodii clustered in the series Villosae but not in the section Ligustrina [33], indicating a weak association between the corolla tube length and the chloroplast genes. All in all, this finding may be due to the weak linkage among these sequences or the molecular markers and corolla tube length traits used in the experiment.

4. With regards to format, the paper needs to be checked thoroughly, including tables and figures, for inconsistencies. For instance, subspecies should not be capitalized; names and words should be written correctly (e.g., “oblata” vs. “oblate”; “Syringa” vs. “Syring”; “section” vs. “Sect”); spaces should be added between words and after punctuations.

Response: We’d like to express our apology to you for the trouble brought to you. We have checked the problems in the format of this paper, and have corrected the writing of Latin names in the manuscript, tables and figures. We have changed the “oblate” in the table and figures to “oblata”, changed “Syring” in Line 282 to “Syringa”, changed the word “Sect.” to “Section” in the entire paper and changed “Ser.” to “Series”. Regarding the problems about capital and small letters of the initial character and italics, I wrote them according to the A reference. The ways of writing are “section Syringa”, “section Ligustrina”, “series Pinnatifoliae”, and “series Pubescentes”.

(A). Li J, Zhang A D. Paraphyletic Syringa (Oleaceae): Evidence from Sequences of Nuclear Ribosomal DNA ITS and ETS Regions. Systematic Botany. 2002; 27(3): 592-597.

5. The paper can also benefit from further editing to improve the English language and to remove some awkwardness throughout the text. For instance, in the Introduction, “disputes about the under the genus classification and interspecific relationships …”, why not just say “the infrageneric classification and relationships”.

Response: We’d like to extend our apology to you for the trouble brought to you. As for the problem of English writing, we have turned to the institutions specialized in English polishing. The proof of polishing is shown in the figure as follows.

6. I deplore the lack of page and line numbers, which makes it more difficult to point to deficiencies accurately.

Response: We’d like to extend our apology to you for the trouble brought to you. Your workload increased due to my negligence. We have added line numbers in the newly submitted manuscript. I’d apologize again for not adding the line numbers.

Once again, thank you very much for your constructive comments and suggestions which would help us both in English and in depth to improve the quality of the paper.

---

## [Editor Report · Decision Letter 1]

29 Jun 2022

PONE-D-22-01886R1Identification and phylogenetic analysis of Syringa based on chloroplast genomic DNA barcodingPLOS ONE

Dear Dr. Shi,

Thank you for submitting your manuscript to PLOS ONE. After careful consideration, we feel that it has merit but does not fully meet PLOS ONE’s publication criteria as it currently stands. Therefore, we invite you to submit a revised version of the manuscript that addresses the points raised during the review process.

The authors have significantly improved the manuscript. Indeed, the term "genus" was placed in front of "*Syringa"* throughout  the text but not in the main title. Please rectify this. Moreover, please remove the short title to avoid possible misunderstandings.

We look forward to receiving your revised manuscript.

Kind regards,

Branislav T. Šiler, Ph.D.

Academic Editor

PLOS ONE
---

## [Author Response · Author response to Decision Letter 1]

3 Jul 2022

Dear Dr. Branislav T. Šiler:

Thank you very much for giving us an opportunity to revise our manuscript. We appreciate the editor and reviewers very much for their constructive comments and suggestions on our manuscript entitled “Identification and phylogenetic analysis of the genus Syringa based on chloroplast genomic DNA barcoding” (ID: PONE-D-22-01886R1).

We have studied reviewers’ comments carefully. According to the reviewers’ detailed suggestions, we have made a careful revision on the original manuscript. All revised portions are marked in red in the “Revised Manuscript with Track Changes” which we would like to submit for your kind consideration.

Kind regards.

Ruihong Yao

E-mail: 337514209@qq.com

Corresponding author : Baosheng Shi

E-mail address: baoshengshi@hebau.edu.cn

Dear Dr. Branislav T. Šiler and reviewers:

Thank you for your letter and the reviewers’ comments on our manuscript entitled “Identification and phylogenetic analysis of the genus Syringa based on chloroplast genomic DNA barcoding” (ID: PONE-D-22-01886R1). Those comments are very helpful for revising and improving our paper, as well as the important guiding significance to other research. We have studied the comments carefully and made corrections which we hope meet with approval. The main corrections are in the manuscript and the responds to the reviewers’ comments are as follows (the replies are highlighted in blue ).

Replies to the Editor Comments:

The authors have significantly improved the manuscript. Indeed, the term “genus” was placed in front of “Syringa” throughout the text but not in the main title. Please rectify this. Moreover, please remove the short title to avoid possible misunderstandings.

Response: I’d like to extend my deep gratitude to you for your suggestions. The words “... of Syringa based on...” in the main title have been changed to “... of the genus Syringa based on...”; “... of Syringa” in line 109 have been changed to “... of the genus Syringa”; The words “... of Syringa ...” in line 194 of the S1 Table title have been changed to “... of the genus Syringa ...”, and the title of S1 Table in supporting information has also been corrected accordingly. In addition, we have deleted the short title.

Replies to the Journal:

Response: Sincerely thanks for your kindly suggestions. We have reviewed the references and there are no retracted papers, and each paper can be searched by hyperlink. In terms of the reference format, we have updated as follows: Reference 14 on lines 370-373 has no page number and volume number (as shown in Fig 1); Reference 16 “Cinnamomum” on line 380 is updated to “Cinnamomum”; the volume issue of Reference 17 on line 385 is 12 (10); the page number of Reference 23 on line 410 is 436; Page numbers pp. 76-84 are supplemented to Reference 24 on lines 411-412, in addition, we have updated the English title of book from “Flora of China” to “Flora Reipublicae Popularis Sinicae”, as well as “Flora of China” on line 98 and line 302 to “Flora Reipublicae Popularis Sinicae”.

Fig 1

Once again, thank you very much for your constructive comments and suggestions which would help us both in English and in depth to improve the quality of the paper.

---

## [Editor Report · Decision Letter 2]

6 Jul 2022

Identification and phylogenetic analysis of the genus Syringa based on chloroplast genomic DNA barcoding

PONE-D-22-01886R2

Dear Dr. Shi,

We’re pleased to inform you that your manuscript has been judged scientifically suitable for publication and will be formally accepted for publication once it meets all outstanding technical requirements.

Kind regards,

Branislav T. Šiler, Ph.D.

Academic Editor

PLOS ONE
---

## [Editor Report · Acceptance letter]

11 Jul 2022

PONE-D-22-01886R2 

Identification and phylogenetic analysis of the genus *Syringa* based on chloroplast genomic DNA barcoding 

Dear Dr. Shi:

I'm pleased to inform you that your manuscript has been deemed suitable for publication in PLOS ONE. Congratulations! Your manuscript is now with our production department. 

Kind regards, 

on behalf of

Dr. Branislav T. Šiler 

Academic Editor

PLOS ONE